# The CALICE SiW ECAL Technological Prototype—Status and Outlook

Roman Pöschl on behalf of the CALICE Collaboration

Laboratoire de Physique des 2 Infinis Irène Joliot-Curie (IJCLab) , Université Paris-Saclay/CNRS/IN2P3, CEDEX, 91405 Orsay, France; roman.poeschl@ijclab.in2p3.fr

**Abstract:** The next generation of collider detectors will make full use of Particle Flow Algorithms, requiring high-precision tracking and full imaging calorimeters. The latter, thanks to granularity improvements by two to three orders of magnitude compared to existing devices, have been developed during the past 15 years by the CALICE collaboration and are now reaching maturity. This contribution will focus on the commissioning of a 15-layer prototype of a highly granular silicon–tungsten electromagnetic calorimeter that comprises 15,360 readout cells. The prototype was exposed in November 2021 and March 2022 to beam tests at DESY and in June 2022 to a beam test at the SPS at CERN. The test at CERN has been carried out in combination with the CALICE Analogue Hadron Calorimeter. The contribution will give a general overview of the prototype and will highlight technical developments necessary for its construction.

**Keywords:** particle flow; calorimeters; high granularity

## 1. Introduction

The design of particle detectors at future high-energy physics experiments and, in particular, at linear colliders is oriented toward the usage of Particle Flow Algorithms (PFAs) for the event reconstruction. These algorithms aim to achieve good jet energy resolution of the order of 3–4% for jet energies between around 45 GeV and several 100 GeV. The algorithms reconstruct individual particles by combining signals in tracking systems and in high-granularity calorimeters [1–3].

The primary objective of the CALICE Collaboration [4] is the development, construction and testing of highly granular hadronic and electromagnetic calorimeters for future particle physics experiments based on the particle flow concept. Experiments at the International Linear Collider (ILC) [5] were the initial goal of the R&D, but the obtained results can be adapted to other proposals for Higgs factories. The collaboration develops common tools such as front-end electronics, digital readout, software and where possible share sizeable mechanical structures. The first stage of this effort is marked by the successful running of so-called physics prototypes that delivered the proof-of-principle that highly granular calorimeters can be constructed and operated [6]. This phase is followed by the development of technological prototypes that aim to address, more than the physics prototypes, the engineering challenges of highly granular calorimeters. The main options in terms of absorber and active material are summarized in Table 1.

This article concentrates on the latest developments for the technological prototype of the SiW ECAL. It will start with a brief overview of this type of calorimeter and a sketch of use cases at future high-energy $e^+e^-$ colliders.

**Table 1.** Overview of calorimeter prototypes developed and tested by CALICE.

| Project | Purpose of Prototype | Absorber | Sensitive Part | Status |
|---------|---------------------|----------|----------------|--------|
| AHCAL | Physics | Stainl. steel/Tungsten | Scintillator | Completed |
| AHCAL | Technological | Stainl. steel | Scintillator | Ongoing |
| TCMT | Physics | Stainl. steel | Scintillator | Completed |
| DHCAL | Physics and Technological | Stainl. steel/Tungsten | RPC Partially GEM | Completed |
| SDHCAL | Physics and Technological | Stainl. steel | GRPC Partially μMegas | Ongoing |
| SiW ECAL | Physics | Tungsten | Silicon | Completed |
| SiW ECAL | Technological | Tungsten | Silicon | Ongoing |
| ScW ECAL | Physics | Tungsten | Scintillator | Completed |
| ScW ECAL | Technological | Tungsten | Scintillator | Ongoing |

## 2. Silicon-Based Calorimeters—Overview and Use Cases

Silicon is particularly well suited for the design of compact and highly segmented calorimeters as required for the particle flow approach. Calorimeters with silicon as the active element have a tradition that goes back to the LEP and SLC era. Small silicon–tungsten calorimeters (with a diameter of around 30 cm) were used e.g., by the OPAL and SLD collaborations for luminosity measurements in the forward regions of the detector [7,8]. This "tradition" will be followed up by the luminosity calorimeter that is designed for future linear electron–positron colliders. A highly segmented calorimeter is also beneficial in more central regions of the detector. A first attempt was made by the ALEPH collaborations. The ALEPH detector featured an electromagnetic calorimeter using subdivided wire-proportional chambers [9]. The vast development of silicon detectors for tracking with ever-increasing surface renders it today possible to envisage silicon-based large surface central calorimeters for future experiments in particle physics. In the past 10–15 years, the R&D on these devices has been conducted within the CALICE collaboration driven by the needs of future linear electron–positron colliders. A proof-of-principle for a silicon–tungsten electromagnetic calrorimeter has been given by the physics prototype that has been operated between 2005 and 2011 [10]. The success of this R&D program inspired the LHC collaboration to consider large-scale calorimeters based on silicon for their high-luminosity upgrades. A prime example is the CMS HGCAL [11,12]. Another LHC upgrade project is the forward calorimeter FOCAL [13,14] of ALICE that combines silicon sensors similar to the ones described in this article with Monolithic Active Pixel Sensors. Please consult Ref. [15] for an extensive overview that goes well beyond this short rśumé.

*Particle Flow and Particle Separation*

The idea behind particle flow is that each particle of the final state is measured in the best suited subdetector. This in turn requires a nearly perfect separation of the final state particles especially in the calorimeters. A typical jet in the final state of high-energy $e^+e^-$ collisions contains:

- 65% charged particles: Up to a momentum of around 100 GeV these are best measured in the tracking system, provided a sufficiently large magnetic field.
- 25% photons: The photons are measured in the electromagnetic calorimeter. The electromagnetic calorimeter has to provide a good photon–hadron separation and has to allow for the proper reconstruction for close-by photons from pion decay. Both become more and more involved with increasing center-of-mass energy.
- 10% neutral hadrons: Here, naturally, the hadron calorimeter is the most relevant device. However, around 50% of the hadrons interact already in the electromagnetic calorimeter.

The use cases presented in the following are taken from optimization studies of the ILD Concept [16], which is a detector proposal for the International Linear Collider that implements the particle flow approach; see also Section 3.

A classical application of particle flow is the separation of W and Z pairs in the reaction $e^+e^- \to W^+W^-/ZZ$ at high energies. The left part of Figure 1 shows this separation for a center-of-mass energy of 1 TeV. The two particles can be clearly separated. However, as also shown in Ref. [16], the separation may become compromised by secondary effects such as imperfect jet clustering but also overlay from $\gamma\gamma$ to hadrons background and missing energy from semi-leptonic heavy-quark decays. The latter two effects can be addressed by improved algorithms that exploit even better the high granularity.

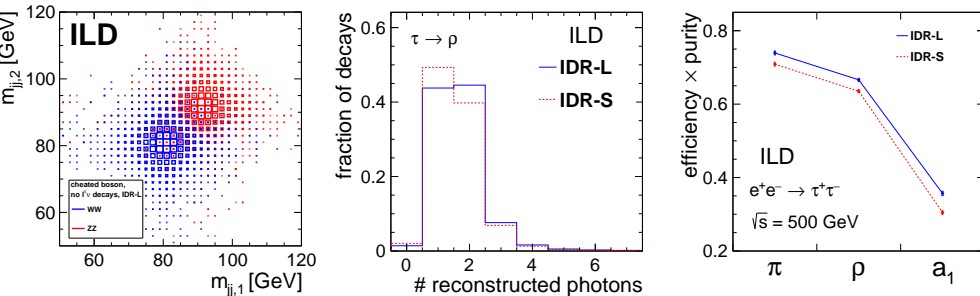

**Figure 1. Left:** Separation of $W^+W^-$ and ZZ final states in full detector simulation of the ILD detector. **Middle:** Efficiency of two photon separation in $e^+e^- \to \tau^+\tau^-$ at $\sqrt{s} = 500$ GeV. **Right:** The ability of distinguishing different $\tau$ decay modes with: $\tau \to \pi^\pm\nu$ ("$\pi$"), $\tau \to \pi^\pm\pi^0\nu$ ("$\rho$"), $\tau \to \pi^\pm\pi^0\pi^0\nu$ ("$a_1$"). The results in the middle and in left figure are shown for two variants of ILD, one with a larger (ILD-L) and one with a smaller inner radius of the electromagnetic calorimeter (ILD-S).

Via spin-correlations $\tau$ leptons are used to determine the CP nature of the Higgs–Boson. Moreover, the measurement of electroweak couplings of the $\tau$ lepton is an important contribution to the search for anomalies. $\tau$ identifiers [17–19] exploit the relatively little populated final state in $\tau$-pair production but rely in particular on the high granularity of the electromagnetic calorimeter. Hadronic $\tau$ decays typically offer the highest sensitivity to the spin, which is due the presence of a single neutron. These decay modes are characterized by the presence of two close-by photons from a neutral $\pi$ decay. The performance of $\tau$ decay mode identification was studied in $\tau^+\tau^-$-pair production events at a center-of-mass energy of 500 GeV [16]. As shown in the middle part of Figure 1, the separation of two photons is successful in 50% of the cases. The product of selection efficiency and purity for the reconstruction of different hadronic decay modes vary between 30% and 75%; see the right part of Figure 1 .

## 3. Highly Granular Silicon Tungsten Electromagnetic Calorimeter for Higgs Factory Detectors

A silicon–tungsten electromagnetic calorimeter is considered for detectors at all Higgs factories. The R&D described in this section is oriented at the electromagnetic calorimeter that is the baseline of the ILD Detector concept. Key parameters of the design of the electromagnetic calorimeters are:

- A sandwich calorimeter with around 30 layers and a depth of around 24 $X_0$ equivalent to 1 $\lambda_I$. The sensitive material is silicon and the absorber material is tungsten. With a ratio of interaction length to radiation length of around nine, tungsten is well suited for an excellent photon–hadron separation, which is an essential ingredient for particle flow detectors. Furthermore, calorimeters have to fit inside the magnetic coil. Therefore, typically, only around 20 cm in depth are available for the calorimeter volume, and tungsten ideally supports a compact design.
- A pixel size of $5 \times 5 \, \text{mm}^2$ as the result of an optimization study carried out in [20].
- A signal-to-noise ratio (SNR) of at least 10. The SNR is defined as the most-probable value of the energy deposited by a minimal ionising particle (MIP) divided by the noise width.

- An electromagnetic energy resolution of around $15–20\%/\sqrt{E} \oplus 1\%$, for the photon measurement.

The left part of Figure 2 is a CAD drawing that illustrates the position of the calorimeters in ILD.

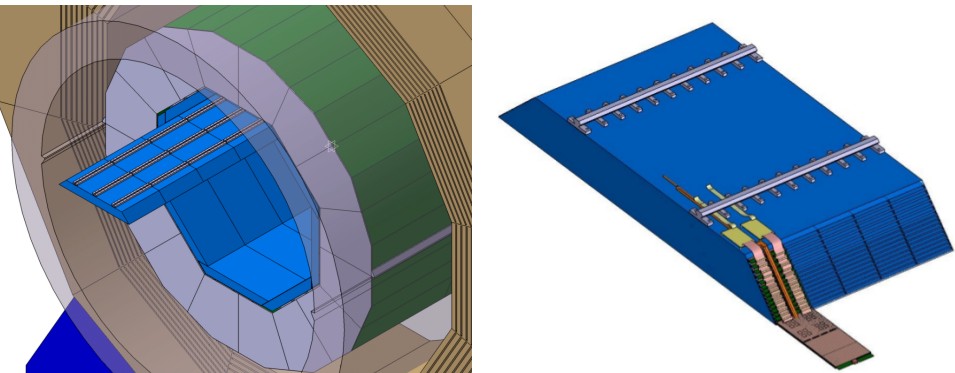

**Figure 2. Left:** A CAD drawing showing the electromagnetic calorimeter of ILD in light blue. The drawing is taken from [20]. **Right:** Drawing of an alveolar structure of a barrel module. The insertion of the slabs into the alveoli is indicated.

The barrel part of the electromagnetic calorimeter is indicated in light blue. The barrel is subdivided into alveolar structures that in turn house the detector layers. In fact, each alveoli houses up to 45 slabs that each consists of two sensitive layers. A drawing of the barrel modules is shown in the right part of Figure 2. The electromagnetic calorimeter is completed by two endcaps [21]. In these endcaps, the alveolar structures have the shape of a quarter. Each layer is subdivided into up to 15 Active Signal Units (ASU). An ASU is the entity of silicon sensors, interface board (PCB) and the readout ASICs. The current lateral dimension of an ASU is $180 \times 180 \, \text{mm}^2$. It is equipped with four silicon sensors processed from 6" wafers. A cross-section through a detector slab is shown in the left part of Figure 3. The overall thickness of a layer with $500 \, \mu\text{m}$ thick silicon sensors without tungsten absorber is between 2.3 and 3.9 mm depending on the overall space needed for the ensemble of front-end electronics and PCB. In ILD, the tungsten thickness per layer is either 2.1 mm or 4.2 mm. An important feature is the highly integrated design. The front-end electronics is embedded into the layer structure. Services such as the digital readout must comply with this compact design. In the following, the R&D aspects to meet the needs of the compact design will be outlined.

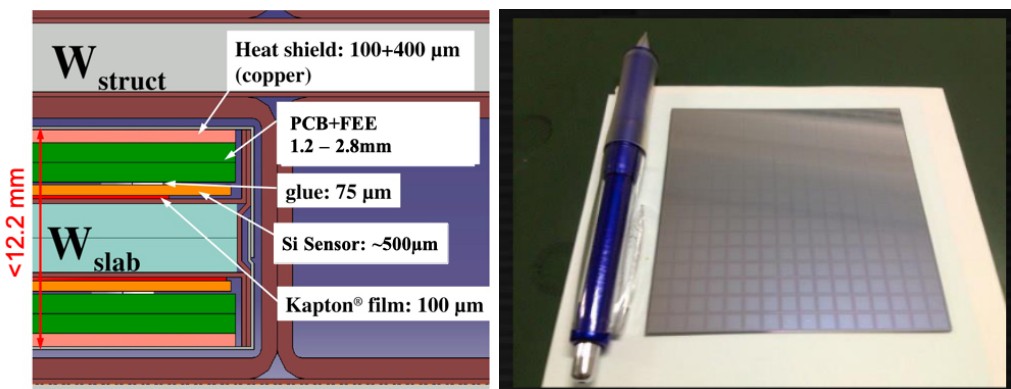

**Figure 3. Left:** Cross-section through a SiW ECAL slab. **Right:** Si sensor, the pads of size $5.5 \times 5.5 \, \text{mm}^2$ are well visible.

### 3.1. Key Elements of an ASU

Silicon Sensors: The right part of Figure 3 shows a silicon sensor that is used for the current prototype. Its lateral all dimension is $90 \times 90 \, \text{mm}^2$ and the sensors are processed from 6" silicon wafers. A sensor is made of n-type silicon with the crystal orientations <100> or <111>. The resistivity of a sensor is typically $5 \, \text{k}\Omega \cdot \text{cm}$. A design requirement is a leakage current of less than a few nA/pixels. However, for cost reasons, also sensors with slightly higher leakage currents are accepted. In recent years, sensors with thicknesses of 320, 500 and 650 μm, respectively, were tested. Thinner sensors lead to a smaller amplitude and require thus a smaller dynamic range of the front-end electronics. Thicker sensors improve the signal-over-noise ration due to the higher amplitude and the smaller sensor capacitance. They may require a larger dynamic range of the front-end electronics. Particular attention was given on the design of the guard ring that surrounds the sensor. A floating guard ring in the so-called physics prototype has lead to signals in pads next to the guard ring, called "square events", due to a capacitive coupling between the pads and the guard ring [22]. Segmented guard rings reduce the number of square events. In the ideal case, the guard ring is omitted, which would increase the sensitive area of a given sensor. For an overview on the tests that have been carried out, see, e.g., Ref. [23]. For the beam tests in 2021 and 2022 at DESY and the CERN (see also below), sensors with either one or no guard ring have been used. Already, earlier tests with sensors produced by HPK in 2011 and 2013 with one guard ring have shown a reduced frequency of square events. The energies at the DESY testbeam facility (1–6 GeV) are too small to observe a sizeable fraction of square events. The analysis of data taken at the CERN SPS testbeam facility (10–150 GeV) in Summer 2022 is ongoing and will allow for a detailed assessment of the effect with state-of-the-art sensors.

Front-end ASICs: The silicon pads are read out by the SKIROC2 ASIC, in its variant SKIROC2a. An image is given in the left part of Figure 4.

A detailed description of the ASIC can be found elsewhere [24]. Here, the main features of the ASIC are recapitulated. The design of the ASIC is oriented at operation at the ILC. At the ILC, the beam comes in bunch trains with a length of around 1 ms per train and a repetition rate of 5–10 Hz. The interbunch distance within a train is 554 or 336 ns. The ASIC is based on SiGe 0.35 μm AMS technology. Its size is $7.5 \times 8.7 \, \text{mm}^2$. It comprises 64 channels with a high integration level that includes variable gain charge amplification and a 12-bit Wilkinson ADC for analogue digital conversion. The ASIC provides a large dynamic range that reaches up to around 2500 MIPS, its low noise ($\sim$1/10 of a MIP) permits auto-triggering at 1/2 MIP. Per channel up to 15 signals can be stored in a Switch Capacitor Array. If no trigger is recorded in a channel of an ASIC, no signal is transmitted to the ADC due to the on-chip zero suppression. The ASIC is designed for low power. In continuous mode, it consumes about 1.5 mW/ch. In power pulsed mode, as available at linear colliders, the power consumption can be reduced to values as low as 25 μW/ch.

Interface Boards (PCB): The PCB design is challenging due to its compactness and the density of channels. The impedance of the PCBs is controlled by design by the CAD tools which permit implementing precisely the rules given by the PCB manufacturers. Applying those methods carefully for our design together with a meticulous care for line impedance adaptation prove that in our case, an expensive post-production control is not required. Furthermore, the design of the lines between the Si pads and the pre-amplifier entries of the ASICs has been optimized for reducing both the parasitic capacitance and all potential sources of crosstalk.

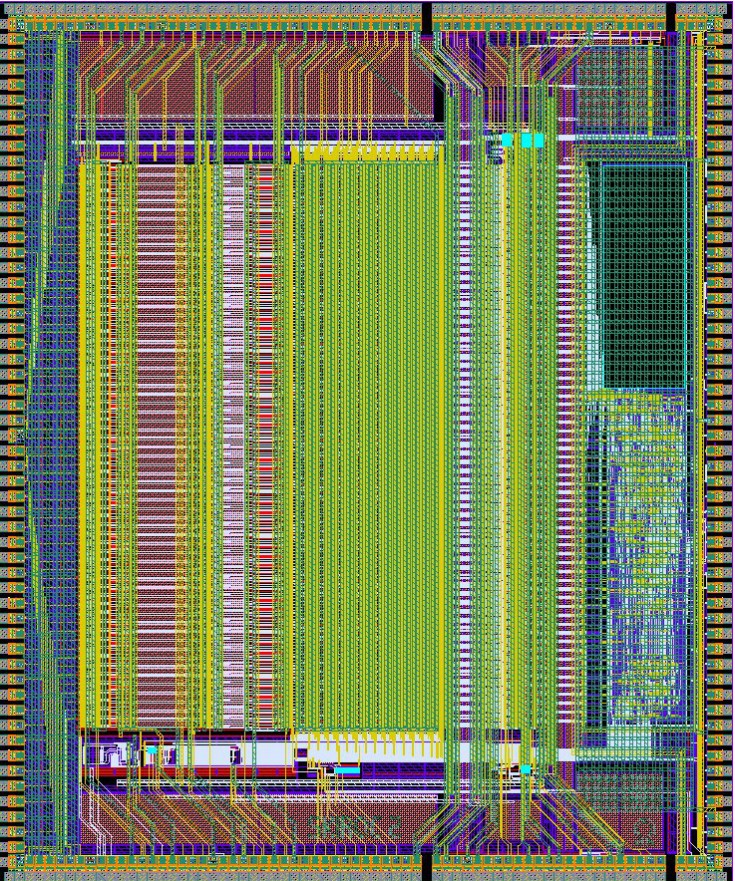

**Figure 4.** Image of the SKIROC2a front-end ASIC.

There exist five PCB versions, three based on BGA-packaged ASICs that have been gradually improved to achieve the minimal required performance. In 2017, an SNR of 12 was reached in the trigger branch (20 in the ADC branch) for BGA-based ASUs in a beam test with a few layers that consisted of one ASU each [25]. Since recently, the fifth version is available, which is thinner than the others (overall 1.2 mm to be compared with around 1.6 mm plus BGA-package height) in which the ASICs are directly wire-bonded onto the PCB. Figure 5 shows three types of PCBs that were used in the 2021 and 2022 beam tests.

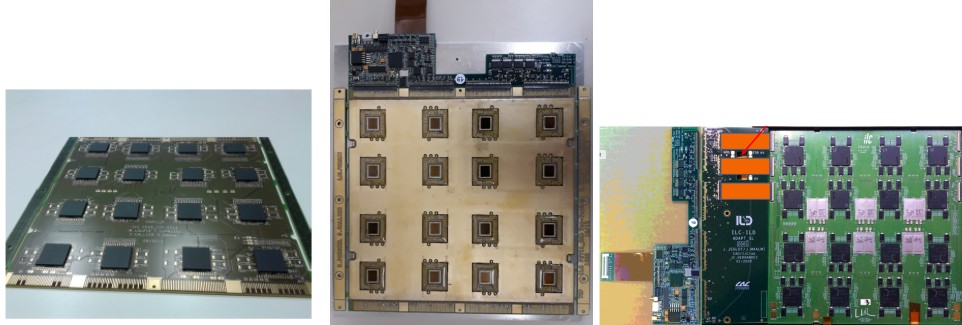

**Figure 5.** PCB variants that were used in the 2021 and 2022 beam tests. In the left and right ones, the ASIC are embedded in BGA packages. They differ by internal routing and external connectivity. The middle one features wire-bonded ASICs in cavities that are sunk into the PCB surface. The ASICS are protected by a transparent potting agent.

Compact readout system:　The system is completed by a compact readout system that became available during 2019 and 2020 [26]. A schematic overview of the readout

chain including the various data links and data bandwidth in given in Figure 6. The slab is connected to an adapter board, called SL-Board (labeled SL-BRD in Figure 6), that provides power regulators and signal buffers and an FPGA to concentrate data to be sent outside. The SL-Board has an overall size of $40 \times 180\,\text{mm}^2$ and is conceived to serve about 10,000 readout cells. The design allows for the connection of a cooling system as described in Ref. [21]. The data are transmitted by a flat kapton cable to a concentrator unit, called the CORE-Module, and readout by the DAQ Computer.

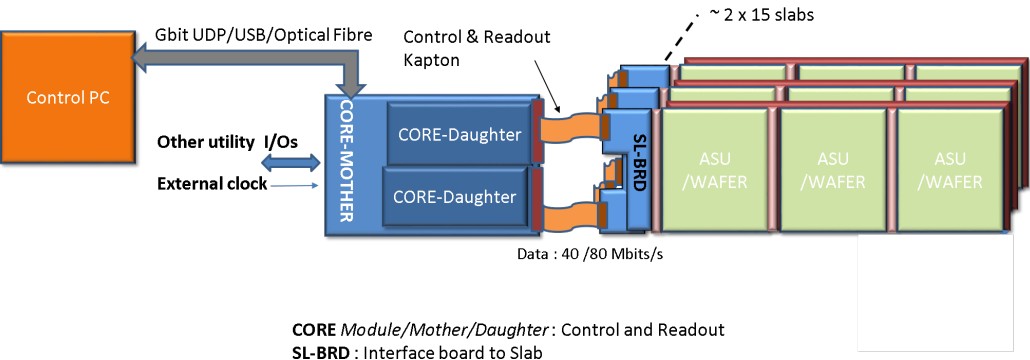

**Figure 6.** Schematic overview of the readout chain for the SiW ECAL prototype.

The acquisition software is written in C-Language and developed under LabWindows CVI. It pilots the communication through the CORE Kapton or through the FTDI Module directly to the SL-Board. It handles the control and readout of a whole detector module consisting of two CORE Kaptons connected to 15 SL-Boards each and up to five ASUs connected in series to each SL-Board. For the recent beam tests, the graphical interface allowed for an extended real-time control of the detector performance.

Further important features are:

- Online Hit Maps and shower profiles that allow for real-time beam and detector tuning, e.g., adaptation of beam rates or thresholds;
- Pedestal measurement and subtraction;
- Charge measurement and histogramming;
- MIP gain correction.

### 3.2. Technological Prototype for 2021/22 Beam Tests

Test beams have been already carried out with earlier versions of the technological prototypes [25]. In these tests, up to seven layers equivalent to 7168 readout cells were used. The readout was made with the predecessor of the current readout system [27]. For 2021, a stack with 15 layers has been compiled. The assembly benefitted from an assembly chain that has been set up during the European project AIDA-2020. The chain comprises (a) silicon sensor tests (b) metrology of PCBs and in-house cabling of the ASUs (c) gluing of the silicon sensors onto the PCBs and (d) the actual assembly into a stack and its commissioning in our workshop before moving to beam tests.

For the beam tests described here, a flexible detector integration has been chosen. As can be seen in Figure 7, the ASUs are mounted on plastic plates and are held in place by 3D-printed plastic rails that also allow for the quick insertion and removal of the tungsten absorber plates. Pictures of the completed fifteen layer stack are shown in Figure 8. The left side shows the stack in the beam test area at DESY. The overall size of the stack, including its mechanical housing, is $640 \times 304 \times 246\,\text{mm}^3$. When being fully equipped with tungsten plates, its weight is around 60 kg. The right picture allows appreciating the high density at the layer extremities that is provided by the compact digital readout system. This is already close to what is expected in a final experiment at a Higgs factory.

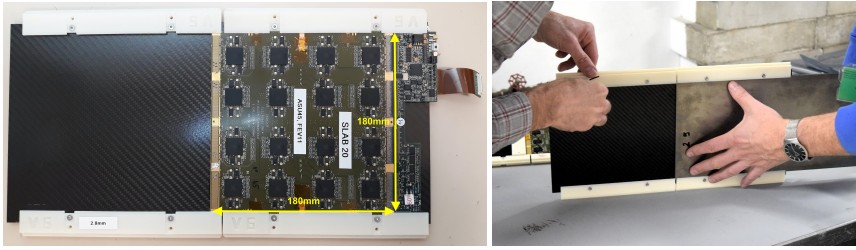

**Figure 7. Left:** Photo of a layer as used for the beam tests described in this article. Visible is the ASU, the SL-Board and the rails that hold in place ASU and tungsten absorber plates. **Right:** Photo of the insertion of the tungsten absorber plates.

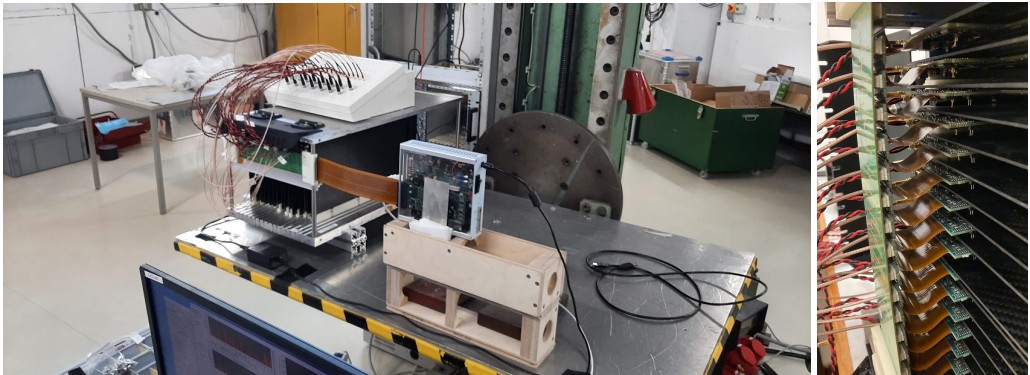

**Figure 8. Left:** Photo of the 15-layer stack of the CALICE SiW ECAL in the beam test area at DESY. Visible are the mechanical housing of the HV and LV cabling that supplies the 15360 cells of the stack as well as the flat kapton cable for data transmission and the CORE-Module. **Right:** Zoom into the extremities of the stack to appreciate the compactness of the ensemble.

The stack has been tested at DESY and at CERN. For the three campaigns (two at DESY, one at CERN), the depth of the stack has been progressively increased in terms of radiation length. In November 2021, the stack was equipped with $10.2\,X_0$, in March 2022 with $15.6\,X_0$ and in June 2022 at CERN with $20.8\,X_0$ of tungsten, respectively. DESY provides electrons in the range of 1–6 GeV. For the detector calibration, the tungsten plates have been removed. In this case, the electrons pass the layers as MIP-like particles. In total, $15 \cdot 1024 \cdot 15 = 230{,}400$ MIP calibration constants (plus pedestals) have to be determined for a given gain setting of the ASIC. This calibration phase includes disabling cells from the auto-trigger in case of a too high noise level. At a more concrete level, an important point is the study of the homogeneity of the layers. A study is presented in Figure 9 for two layers with data taken at DESY. The black regions signal cells that were not available for the analysis, since they were either disabled or found to be not responsive. If a cell is present, the efficiency is typically well above 90%. The black regions in the left part of Figure 9 are exclusively cells for which the trigger has been disabled. The reason are routing issues in the PCB that will be corrected in future versions. For the layer on the right-hand side, it was found that many cells did not response to energy depositions by particles. The reason will be investigated with high priority in the coming months. Most likely, the sensor delaminated from the PCB in the corresponding regions. We were already able to spot on a test bench failing cells by measuring in situ the DC output voltage of the preamplifier of the corresponding channels ("analogue probes"), which is a possibility that became available during 2022. At CERN, the prototype was exposed to μ, π and electron beams in the energy range between 10 and 150 GeV. The data analysis is ongoing. As first impression, Figure 10 shows event displays in which one and two electrons hit the detector during the beam test at CERN. Note for completeness that the beam test at CERN has been carried out in combination with the technological prototype of the CALICE AHCAL. For this combined running, both the SiW ECAL and the AHCAL have been implemented into the EUDAQ2 Data Acquisition suite [28].

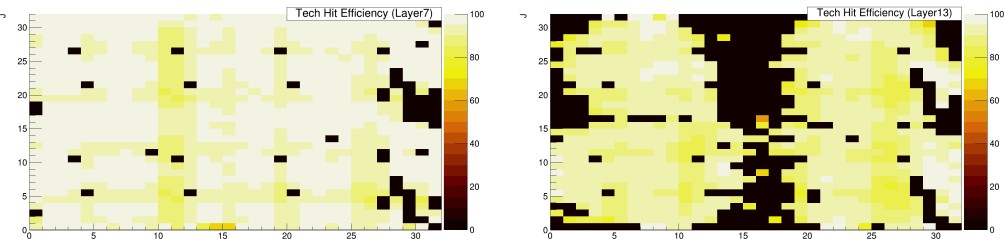

**Figure 9. Left:** Example for a layer with nearly complete acceptance. **Right:** Example for a layer with reduced acceptance mainly due to ineffieciencies in the response to energy depositions. Here, *I*, *J* indicate the pad index and the color code represents the efficiency of a pad to energy depositions by MIP-like particles.

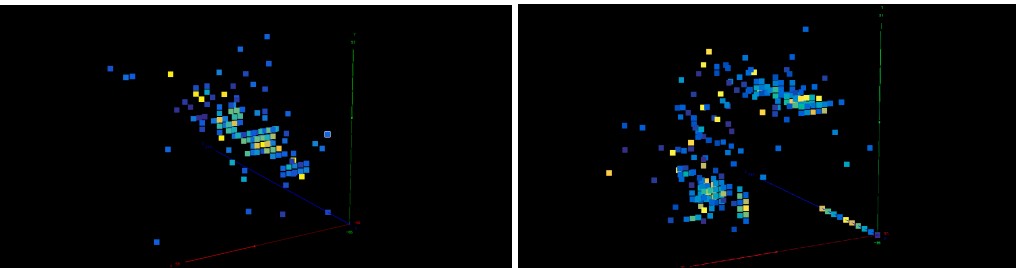

**Figure 10. Left:** A 20 GeV electron recorded in the SiW ECAL. **Right:** Two 20 GeV electrons in the SiW ECAL. The distance between the two showers is about 5 cm.

The construction and operation of the technological prototype is accompanied by its implementation into simulation. The geometry is described in the DD4HEP framework [29] as an interface to the GEANT4 toolkit [30]. The simulation will take the correct digitisation of the signal into account. Therefore, the response function of the fast and slow shapers are implemented into the signal processing chain in the simulation.

### 3.3. Next Steps

The stack described before forms the basis for the continuation of the R&D. In the following, the most important items are briefly sketched.

### 3.3.1. Development of a Power Pulsing System of the Detector

A particular feature of detectors at linear colliders is that the detectors can be power pulsed. For this, ASUs are under development that allow for storing the power {locally, i.e., next to the ASICs. This design avoids large peak currents along the layers that will feature around 10,000 cells and will be between 1.5 and 2.0 m long. The first tests carried out in Winter 2021/2022 were encouraging in terms of the noise of the detector. We plan to produce between 15 and 20 ASUs with the same size as before, i.e., $180 \times 180 \, \text{mm}^2$. These ASUs will be assembled into a stack for beam test measurements in the coming years. On the other hand, they will be used to continue the R&D on real size layers, i.e., chains of up to 12 ASUs in continuation of the work that has been published in Ref. [31]. Note in passing that in contrast to the previous design, the ASUs will be more autonomous. The bias voltage will be transmitted through the interconnection of the ASUs and supplied to the sensors of the individual ASUs. This avoids the design of long copper/kapton HV sheets that are difficult to handle and bear the risk that a damaged sheet compromises an entire layer.

### 3.3.2. Timing for Highly Granular Calorimeters at Higgs Factories

A precise measurement of the time structure of electromagnetic and in particular of hadronic showers can improve significantly the quality of the event reconstruction in high-energy particle collisions. In fact, different processes have different time scales. This

knowledge can be exploited for the correction of the visible energy and possibly of the positioning. The potential of an excellent time resolution on the energy measurement in highly granular calorimeters has been nicely demonstrated in Refs. [32,33]. The requirements on timing precision for a calorimeter system at Higgs factories will be determined. Ideally, the time resolution becomes comparable to or better than the calorimeter cell size divided by the speed of light (1 cm = 30 ps × c). This would enable the detector to follow the particle shower development with similar resolution in both time and space, allowing to impose causality constraints in the particle flow analysis. Once the performance goals for the calorimeter timing precision are known, the requirements will have to be translated into a conceptual design of the calorimeter and its electronics.

### 3.3.3. R&D on Power Economic Solutions

The tendency for future Higgs factories is an increase of the beam collision frequency compared to the case of the International Linear Collider. For a comprehensive overview, see [34]. For example, at circular $e^+e^-$ colliders as the FCCee, the envisaged bunch distance is around 35 ns at the Z pole and around 1 us for HZ running. The continuous beam will not allow for the application of power pulsing. The actual data rate will also depend on the corresponding cross-sections and angular distributions of the relevant physics processes. However, also for linear colliders, higher collision frequencies are envisaged and have to be taken into account in the medium and long-term planning of the R&D. To this, we add that an improved timing resolution will yield an increase of the power consumption of the front-end electronics. The electronics have to minimize the need of cooling in order to not compromise the quality of the PFA. The goal should be to keep the power consumption well below 1 mW/channel. This may be achieved with smaller feature sizes of the components of the front-end electronics. This, however, may lead to a penalty on the dynamic range of the electronics. In addition, the compactness of the readout electronics must remain at the same level as today while being able to cope with significantly increased data fluxes. The R&D in the wider sense has to be carried in close coordination with the R&D on cooling systems that may become unavoidable in case of high collision frequencies. A full system study in close coordination with detector optimisation studies with relevant physics processes has to be carried out.

### 3.3.4. R&D on Silicon Sensors

The general trend is to produce silicon sensors from wafers larger than 6″. The CMS Collaboration uses sensors produced from 8″ wafers for the construction of the CMS HGCAL. The R&D for a silicon–tungsten calorimeter for a future Higgs factory will naturally benefit from the CMS HGCAL in terms of the availability of a production line for 8″ sensors at HPK and experience with testing and handling large sensors. A difference may be the thickness. For the purpose of radiation hardness, CMS uses sensors with a maximal thickness of 300 μm. This requires thinning of the sensors from the standard thickness of 725 μm for sensors processed from 8″ wafers. At an $e^+e^-$ collider, one can work with thicker sensors, which facilitates the production and the handling. Another difference is the shape of the sensors. CMS uses hexagonal sensors, while the design of the SiW ECAL is based on quadratic or at least rectangular sensors.

## 4. Summary and Conclusions

This article described the path toward a successful operation of a fifteen-layer stack of a highly granular silicon tungsten electromagnetic calorimeter in beam tests at DESY and CERN in 2021 and 2022. These tests constitute a major milestone for technological prototype, in particular concerning the performance of the compact readout system. The recorded data are a rich data set to study the detector performance allowing for spotting strong but also weak points of the current R&D. A first impression indicates the particle separation power but also inhomogeneities in the detector response that will have to be understood further. In total, the current technological prototype represents a powerful

infrastructure to conduct conclusive system tests now and in the coming years. A new type of ASU will allow for finalizing the R&D in terms of power pulsing and for bringing us to the "eve" of an engineering prototype in around the next two years. It is important to understand the benefit of high-precision timing. An important aspect of the future R&D is the development and testing of low-power electronics and readout systems that can cope with the increased data flux to be expected at future $e^+e^-$ colliders.

As a final remark, it is worth pointing out that highly granular calorimeters may also serve at smaller experiments before the actual construction of a large-size calorimeters with the corresponding long lead time. An example for such a smaller project is the LUXE Experiment at DESY/Hamburg [35].

**Funding:** This project has received funding from the European Union's Horizon 2020 research and innovation programme under grant agreement No 101004761 .

**Data Availability Statement:** Not applicable.

**Acknowledgments:** R.P. would like to thank the organisers of the CALOR 2022 for the perfect organisation and the hospitality at Brighton. The measurements leading to these results have been performed at the Test Beam Facility at DESY Hamburg (Germany), a member of the Helmholtz Association (HGF) and at the SPS Test Beam Facility of the European Center for Nuclear Research (CERN). R.P. would also like to thank Yuichi Okugawa (IJCLab) and Hector Garcìa (CIEMAT) for having provided the plots for Figures 9 and 10. Finally, R.P. would like to thank Dominique Breton. Jihane Maalmi and Jérôme Nanni for having helped me with technical questions in the course of the revision of the manuscript.

**Conflicts of Interest:** The authors declare no conflict of interest.

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
