# Peer review of "The CALICE SiW ECAL Technological Prototype—Status and Outlook"

_instruments, doi:10.3390/instruments6040075_

Round 1
Reviewer 1 Report
line 6 --- comprises
line 11 - remove the round bracket on the right
line 28 - put of instead of "on"
line 43 - is missing a hyphen in "electron positron"
line 63 -- remove capital letter from "Pairs"
line 76 - dot instead of comma --- a neutral pion decay,
line 80 - missing the dot -- right part of Fig. 1.
line 97 - missing a dot--as being optimal.
line 108 - missing "of" ---left part of Fig. 3.
line 132 - a typo mistake---in 2011 and 2013
line 145 - Fig. 4 instead of "Figure 4", to keep the manuscript consistent.
line 150 - typo mistake---ASIC is based
In Figure 7 ---keep the same size for both images.
line 209 - Fig. 8 instead of "Figure 8", to keep the manuscript consistent.
line 286 --missing a dot at the and of the sentence "...in the coming years."
general comment for References: for all arXiv the link doesn't work.
Author Response
line 6 --- comprises
---> done
line 11 - remove the round bracket on the right
---> done
line 28 - put of instead of "on"
---> done
line 43 - is missing a hyphen in "electron positron"
---> done
line 63 -- remove capital letter from "Pairs"
---> done
line 76 - dot instead of comma --- a neutral pion decay,
---> done
line 80 - missing the dot -- right part of Fig. 1.
---> done
line 97 - missing a dot--as being optimal.
---> done and slightly rephrased.
line 108 - missing "of" ---left part of Fig. 3.
---> done
line 132 - a typo mistake---in 2011 and 2013
---> done
line 145 - Fig. 4 instead of "Figure 4", to keep the manuscript consistent.
---> Here it is at the beginning of a sentence. Therefore I believe that it's good as it is.
line 150 - typo mistake---ASIC is based
---> done
In Figure 7 ---keep the same size for both images.
---> done
line 209 - Fig. 8 instead of "Figure 8", to keep the manuscript consistent.
---> done
line 286 --missing a dot at the and of the sentence "...in the coming years."
---> done
general comment for References: for all arXiv the link doesn't work.
---> Hmmm ... when I compile the draft on my machine the links do work. I will see with the journal whether there is a technical problem somewhere.
Author Response
Dear Reviewer,
thanks a lot for the review and the encouraging comments.I agree that more results from the beam tests would have been nice. The analysis is at full swing and the article represents the status of September 2022.
Please find below my replies to your comments.
All the best,
Roman
l 46: add “the” in front of forward calorimeter
---> done
l 54: remove “of” after 65%. Same for lines 56 and 60
---> done
l 76: change , after decay to .
---> done
Fig 1: There are inserts labelled with IDR-S and IDR-L. It is not clear what this stands for and why there are two lines. This needs to be explained in the text or caption.
---> explained now in the caption. ILD-L means Large version of ILD and ILD-S means Small version of ILD.
l 133: The energies at DESY → The energies at the DESY test beam
---> done
Fig 4, caption: 2nd line ASIC → ASICs ; 3rd line: The middle one feature wire bonded → The middle PCB features wire bonded
---> done
l 150: based
---> Hmmm ... cannot spot this comment. Next to line 150 there is well the word "based" in "The ASIC is based on SiGe ..."
l 226: forms
---> done
l 240: are
---> done
Ref 11: Is there a link to the LoI? Would be good to add.
---> Well, the LoI is only on the CERN CDS server and it seems that the style file of the journal doesn't accept url entries (only doi and arXiv). The solution is that I added the Snowmass contribution that is on the arXiv.
Ref 15: This is only a preprint. Is there any publication for this?
---> No, there is no aper publication for this.
Ref 21: cite NIM A explicitly
---> done
Ref 29: This is only a preprint. Is there any publication for this?
As far as I can tell there is still no journal paper for this pre-print.
Reviewer 3 Report
Congratulation for a well written and interesting paper.
I would like to ask a clarification about
209-211 and figure 8 right. It looks like the not responsive fraction of the detector is sizeable here. Did you understand the reason for that. I understood from line 185 that the detector was commissioned in the lab. Were the dead channel already present there?
Please find a few minor comments here below
11: extra ")" left
16: Here and through the paper I suggest to substitute around with about, but this is most matter of taste
44: collaboration -> collaborations
57: photon hadron -> photon-hadron
63: Pairs -> pairs
66: add a coma after the Ref [12]
76: decay, -> decay.
caption of figur2 4: feature -> features
150: baed -> based
171: Further ... may go to a new line
222: GEATN4 -> GEANT4
234: AUS-> ASU
240 that a -> that is
Author Response
Dear Reviewer,
thanks a lot for the review and the encouraging statements. Please find my replies below.
Cheers,
Roman
209-211 and figure 8 right. It looks like the not responsive fraction of the detector is sizeable here. Did you understand the reason for that. I understood from line 185 that the detector was commissioned in the lab. Were the dead channel already present there?
Yes the detector was commissioned in the lab to the level that we could set already reasonable noise thresholds to have a quick start at the beam tests. We had also some indications of inefficiencies with cosmics (and by taking a closer look at data recorded earlier). However, to see the full pattern of inefficient regions we had to take high statistics runs in cosmics. The investigation for the reason of the sensor delimitation is at full swing. We have some ideas but no real explanation at this point. For the proceedings I decided to play with open cards and mention the problem. At the recent CALICE Meeting I have learned that understanding the problem will be of general importance. That's all I can say at this point.
11: extra ")" left
---> done
16: Here and through the paper I suggest to substitute around with about, but this is most matter of taste
44: collaboration -> collaborations
---> done
57: photon hadron -> photon-hadron
--> done
63: Pairs -> pairs
---> done
66: add a coma after the Ref [12]
---> done
76: decay, -> decay.
---> done
caption of figure 4: feature -> features
---> done
150: baed -> based
--> done
171: Further ... may go to a new line
---> done
222: GEATN4 -> GEANT4
---> done
234: AUS-> ASU
---> done
240 that a -> that is
---> done but "that are" is correct.
Reviewer 4 Report
The core of this article is an R&D status report on a high-granularity SiW calorimeter prototype for potential application in future collider detectors, including recent beam tests at SPS and DESY. This represents potentially interesting information for the collider physics community.
That said, I think that the article as written is unclear in many places, including weakly organized descriptions of the design concept prototype and tests, which detract from its potential impact. My comments below include suggestions of places where the writing can be strengthened. I hope they will be useful.
-----------------
1. Introduction
This introduction is supposed to present the context for the SiW ECAL technology prototype within the context of CALICE. But a reader not already familiar with CALICE is directed to the collaboration Twiki (ref [4]) or a 73-page status report from 2012 [5] to understand the projects named in Table 1 (AHCAL, TCMT, DHCAL...). Since the rest of the paper focuses on a SiW ECAL anyway, I think it would be more useful to introduce CALICE in terms of the purpose, common developments (including FE and DAQ) and prototype development with different absorbers/active layers.
It might also be good to mention some of the near-term and longer-term applications being targeted. ILD simulation studies are already being presented in Figure 1 without prior mention of the detector.
2. Silicon based calorimeters
"Tradition" is not a particularly good motivation for a choice of technology. At the very least, you should present your assertion that silicon is "particularly well suited" for PFA-oriented calorimeter designs. Referencing a review article on Si-based calorimetry would be good as well. For example, the following article provides a quite extensive review of past and current Si calorimeter development, including CALICE and ILD:
Silicon Calorimeters
J.-C. Brient, R. Rusack, F. Sefkow
Annual Review of Nuclear and Particle Science 2018 68:1, 271-290
Line 84: It would be good to mention the "proof of principle" physics prototype calorimeter already in Section 2.
Line 85: The "R&D described in the following" seems to be a description of the ILD ECAL design concept, which has an unclear relationship with the actual technological prototype that (I think) begins to be described in section 3.1. Please make a clear boundary between discussion of the ILD design and the prototype R&D work.
Line 87: The following bulleted list and paragraph contain a series of loosely-connected statements with unclear ordering, and it is difficult to understand what is important/relevant. The section (and many of the following ones) would benefit from trimming less important details and presenting the important ones in a better-organized text, including motivations for the design choices, etc.
Line 124: Sensors with 320, 500 and 650um thick were tested, with ~500um indicated in the ILD design concept (Figure 3). What was the motivation for choosing this thickness? Also, how does this relate with the "standard thickness" of 725um mentioned in section 3.3.4?
Line 133: You haven't described the 2021-2022 tests at DESY and CERN yet, so at the minimum you should mention the DESY and SPS beam energies to show why you are waiting for the SPS 2022 data analysis.
Line 137 (Interface Boards): Besides photographs, can you provide any more details on the PCBs, such as number of layers, impedance control, or strategies for reducing parasitic capacitance?
Line 156: The front-end ASICs are designed for low power, less than 1 mW/channel in continuous mode. What clock frequencies are being used? This is relevant to section 3.3.3, which correctly points out that power consumption increases with clock frequencies, which in turn generally increase with higher collision rates.
Line 159 (Compact readout system): In addition to (or instead of) the photos in Figure 5, a control/dataflow diagram of the compact readout system would be very useful for understanding the front-end and readout functionality and organisation.
Line 167: It is mentioned here that the acquisition system was developed under LabWindows CVI. Is this the basis for the "graphical interface" mentioned in line 170?
Section 3.2 (Technological prototype for 2021/22 beam tests)
In this section, information about the prototype, DESY tests, CERN tests, calibration and simulation are combined together in no obvious order. It would be much clearer to present first the SiW ECAL stack(s) used in the runs, followed by the DESY and CERN activities (and results) in separate sections.
Line 223: Is there a figure available for the shaper response functions implemented in the signal processing simulation? Have they been tuned with test beam data?
Line 238: In what way will the new ASU design be "autonomous"? Does this apply only to power distribution, or other functionality as well?
Section 3.3.2 (Timing): Instead of simply referring the reader to two other articles, can you provide some representative numbers/figures to give the reader an idea of what kinds of timing requirements are being considered? Otherwise this section is not very informative.
Section 3.3.3 (power economical solutions): As in the previous subsection, some numbers here would be very useful. The current ILD power consumption limit of 1 mW/channel depends on the collision frequency (which has not been reported in this paper). It would also help to mention some of the contemplated frequencies of other future Higgs factories (for example see https://doi.org/10.48550/arXiv.2209.05827).
Section 3.3.4 (R&D on silicon sensors)
In what ways do you propose to "capitalise on the CMS experience". Would this mean testing CMS wafers (with hexagonal sensors and 320um thickness) in the SiW ECAL prototype?
Author Response
Dear Referee,
thank your very much for the careful reading of the manuscript. Please find below the replies to your comments.
- Introduction
This introduction is supposed to present the context for the SiW ECAL technology prototype within the context of CALICE. But a reader not already familiar with CALICE is directed to the collaboration Twiki (ref [4]) or a 73-page status report from 2012 [5] to understand the projects named in Table 1 (AHCAL, TCMT, DHCAL...). Since the rest of the paper focuses on a SiW ECAL anyway, I think it would be more useful to introduce CALICE in terms of the purpose, common developments (including FE and DAQ) and prototype development with different absorbers/active layers.
---> I have added a few words along the lines suggested by the referee. Beyond I think that the table is adequate to sketch the different options in CALICE.
It might also be good to mention some of the near-term and longer-term applications being targeted. ILD simulation studies are already being presented in Figure 1 without prior mention of the detector.
--> ILD now introduced before Fig. 1. Some forward referencing seems to be ok for me since after all the article is rather short.
2. Silicon based calorimeters
"Tradition" is not a particularly good motivation for a choice of technology.
---> I think here the text is clear at it is.
At the very least, you should present your assertion that silicon is "particularly well suited" for PFA-oriented calorimeter designs.
---> Done.
Referencing a review article on Si-based calorimetry would be good as well. For example, the following article provides a quite extensive review of past and current Si calorimeter development, including CALICE and ILD:
Silicon Calorimeters
J.-C. Brient, R. Rusack, F. Sefkow
Annual Review of Nuclear and Particle Science 2018 68:1, 271-290
---> Article now cited.
Line 84: It would be good to mention the "proof of principle" physics prototype calorimeter already in Section 2.
---> Done.
Line 85: The "R&D described in the following" seems to be a description of the ILD ECAL design concept, which has an unclear relationship with the actual technological prototype that (I think) begins to be described in section 3.1. Please make a clear boundary between discussion of the ILD design and the prototype R&D work.
---> Honestly, I think that here the text was/is clear as it is.
Line 87: The following bulleted list and paragraph contain a series of loosely-connected statements with unclear ordering, and it is difficult to understand what is important/relevant. The section (and many of the following ones) would benefit from trimming less important details and presenting the important ones in a better-organized text, including motivations for the design choices, etc.
---> All statements (design parameters) are important. Still, I have revised the bulleted list.
Line 124: Sensors with 320, 500 and 650um thick were tested, with ~500um indicated in the ILD design concept (Figure 3). What was the motivation for choosing this thickness? Also, how does this relate with the "standard thickness" of 725um mentioned in section 3.3.4?
---> It doesn't seem unusual to me to test different sensor thicknesses in a long R&D program. The sensor thickness is always a trade-off between signal amplitude (to limit the necessary dynamic range of the front-end electronics) and Signal-over-Noise ratio at MIP level. Thinner sensors have a smaller amplitude but also a higher capacitance. Thicker sensors have a smaller capacitance but a higher amplitude. The thickness mentioned in 3.3.4 is the standard thickness for sensors processed from 8" wafers. That is at least what HPK would propose to us. For reasons of radiation hardness CMS-HGCAL has to work with thinner sensors. A an e+e- collider radiation hardness would not be an issue.
Line 133: You haven't described the 2021-2022 tests at DESY and CERN yet, so at the minimum you should mention the DESY and SPS beam energies to show why you are waiting for the SPS 2022 data analysis.
---> Beam energies now mentioned. Beyond, I still think that the argument fits best here, even if the beam tests themselves are only mentioned later on.
Line 137 (Interface Boards): Besides photographs, can you provide any more details on the PCBs, such as number of layers, impedance control, or strategies for reducing parasitic capacitance?
---> Done (in coordination with our engineers).
Line 156: The front-end ASICs are designed for low power, less than 1 mW/channel in continuous mode. What clock frequencies are being used? This is relevant to section 3.3.3, which correctly points out that power consumption increases with clock frequencies, which in turn generally increase with higher collision rates.
---> The clock frequency of the ASIC that we use is 5 MHz derived from a 40 MHz external clock. That's however not the main point. It's indeed rather the collision rate or more general the beam structure. A higher collision rate may call for a faster shaping time in order to avoid pile-up. A continuous beam calls for a different digitisation and acquisition chain. In the SKIROC2 ASIC the power consumption is dominated by the power consumption of the preamplifier (roughly 90% of the entire power consumption). In the HGCROC ASIC (for the CMS HGCAL) the power consumption is roughly equally shared between the analogue part and the digital part.
I added the beam parameters of the ILC and corrected a mistake on the actual power consumption.
Line 159 (Compact readout system): In addition to (or instead of) the photos in Figure 5, a control/dataflow diagram of the compact readout system would be very useful for understanding the front-end and readout functionality and organisation.
---> A schematic view on the entire chain is added (again in coordination with the corresponding engineers).
Line 167: It is mentioned here that the acquisition system was developed under LabWindows CVI. Is this the basis for the "graphical interface" mentioned in line 170?
---> Yes, it's part of it.
Section 3.2 (Technological prototype for 2021/22 beam tests)
In this section, information about the prototype, DESY tests, CERN tests, calibration and simulation are combined together in no obvious order. It would be much clearer to present first the SiW ECAL stack(s) used in the runs, followed by the DESY and CERN activities (and results) in separate sections.
---> Text reordered. For convienience two photos of an individual layer have been added.
Line 223: Is there a figure available for the shaper response functions implemented in the signal processing simulation? Have they been tuned with test beam data?
---> I think that at this point it is sufficient to mention that this is ongoing. I agree that this will be more relevant once we publish physics results of the beam test data. If you like to the tuning is ongoing although the initial implementation is based on test bench measurements by the engineer that has designed the ASIC. I have rephrased this part a little bit in order to "damp" the expectation of a reader (Still, I would like to honour the ongoing work of one of our postdocs).
Line 238: In what way will the new ASU design be "autonomous"? Does this apply only to power distribution, or other functionality as well?
---> It applies to the distribution of the bias voltage for the sensors, but this is important. Before the bias voltage was transmitted over long kapton sheets to which the ASUs were glued. It would have been very difficult to exchange an ASU.
Section 3.3.2 (Timing): Instead of simply referring the reader to two other articles, can you provide some representative numbers/figures to give the reader an idea of what kinds of timing requirements are being considered? Otherwise this section is not very informative.
--> I give now a kind of ultimate goal (knowing that this is very difficult to achieve at system level).
Section 3.3.3 (power economical solutions): As in the previous subsection, some numbers here would be very useful. The current ILD power consumption limit of 1 mW/channel depends on the collision frequency (which has not been reported in this paper). It would also help to mention some of the contemplated frequencies of other future Higgs factories (for example see https://doi.org/10.48550/arXiv.2209.05827).
---> Thanks. I cite the paper and give some numbers for the FCCee.
Section 3.3.4 (R&D on silicon sensors)
In what ways do you propose to "capitalise on the CMS experience". Would this mean testing CMS wafers (with hexagonal sensors and 320um thickness) in the SiW ECAL prototype?
---> "Benefit" is maybe the better wording. On the other hand it seems to be clear that one project can profit from the experience of another project that uses 8" sensors. Even if we don't have the same thickness we would benefit from the fact that there is an 8" production line at HPK. Further, there are test stations available at CERN for the CMS 8" sensors (that may need to be adapted for the geometry but still). These test stations are operated by colleagues that are also in CALICE. I know that the diffusion process for the doping may be different for 8" wafers than for 6" wafers. The CMS sensors suffered from a backplane fragility that requires an utmost careful handling.
The proceedings are not the place to dwell on all this but latest after the rewording of the section, it should be obvious that the CMS work on the 8" is useful also for us.
Reviewer 5 Report
The manuscript describes a highly granular silicon-tungsten electromagnetic calorimeter (SiW-ECAL) and focuses on the concept and electronic readout. The study follows a design for the ECAL of the International Large Detector (ILD) concept, but could be applied to other possible high-energy detectors at a future Higgs factory.
The content of the manuscript is very interesting. The description of the calorimeter concept is followed by results from beam-test studies and Monte-Carlo simulations. Everything looks scientifically very solid.
The manuscript is exceptionally well written for a conference contribution. It would qualify for a stand-alone original research article. I don't see any scientific flaw and would recommend the manuscript for publication without further corrections. The only correction I ask for is to remove Fig.5 (right) which seems to have low resolution and, in addition, has been published in Ref. 23.
Author Response
Dear Referee,
thanks for the review and for the positive assessment of the draft. Please find below the reply to your comments.
Cheers,
Roman
The only correction I ask for is to remove Fig.5 (right) which seems to have low resolution and, in addition, has been published in Ref. 23.
---> Following a suggestion of another review I have replaces this figure by a figure (cartoon) that indicates the full readout chain.